# Physical Exercise Effects on Cardiovascular Autonomic Modulation in Postmenopausal Women—A Systematic Review and Meta-Analysis

**DOI:** 10.3390/ijerph20032207

**Published:** 2023-01-26

**Authors:** Juan Carlos Sánchez-Delgado, Adriana Marcela Jácome-Hortúa, Kelly Yoshida de Melo, Bruno Augusto Aguilar, Stella Vieira Philbois, Hugo Celso Dutra de Souza

**Affiliations:** 1Grupo de Investigación Ser Cultura y Movimiento, Facultad de Salud, Universidad Santo Tomás-Bucaramanga, Santander 680001, Colombia; 2Universidad de Santander, Facultad de Ciencias Médicas y de la Salud, Bucaramanga 680003, Colombia; 3Laboratory of Physiology and Cardiovascular Physioterapy, Ribeirão Preto Medical School, University of São Paulo, Ribeirão Preto 14049-900, Brazil

**Keywords:** autonomic nervous system, menopause, physical activity, secondary prevention

## Abstract

Background: The cardioprotective effect of physical exercise has been demonstrated in several studies. However, no systematic or updated analysis has described the effects of physical exercise on cardiovascular autonomic modulation in postmenopausal women. Aim: to describe the effects of physical exercise on cardiovascular autonomic modulation in postmenopausal women. Methods: The Scopus, PubMed, and Embase databases were searched for randomized clinical trials published between January 2011 and December 2021, and regarding the effects of physical exercise on cardiovascular autonomic modulation in postmenopausal women. Two independent authors processed the citations. The methodological quality was evaluated using the PEDRo scale. Results: Of the 91 studies identified, only 8 met the inclusion criteria, of which 7 had fair or poor methodological quality. The analyzed studies investigated the effects of functional training, whole-body vibration, muscular resistance, stretching, and aerobic exercises performed at home or at the gym. The majority of these exercise modalities showed improvements in heart-rate variability (HRV) indices and in the low-frequency band of blood pressure variability. The meta-analysis shows that exercise increased the standard deviation of instantaneous beat-to-beat variability (SD1) (mean difference (MD) = 3.99; 95% confidence interval (CI) = 1.22 to 6.77, n = 46; I^2^: 0%) and the standard deviation of long-term variability (SD2) (MD = 11.37; 95% CI = 2.99 to 19.75; n = 46; I^2^: 0%). Conclusions: Aerobic exercise and some nonconventional training modalities may have beneficial effects on cardiovascular autonomic modulation in postmenopausal women. More high-quality studies are still needed to further confirm their efficacy and safety.

## 1. Introduction

Menopause is characterized by ovarian hormone privation, which is often followed by dysfunction in cardiovascular autonomic modulation. These alterations involve an increase in sympathetic influence on the heart and vascular beds, and a decrease in vagal influence on the heart. Both factors are associated with increased cardiovascular risk and mortality in this population [1,2,3,4]. The reduction in heart-rate variability (HRV) and baroreflex sensitivity (BRS) are physiological parameters that frequently reflect these dysfunctions and the increase in blood-pressure variability (BPV). These indicators show a low capacity of the heart to respond to multiple physiological and environmental stimuli, including compensation for disorders induced by the same diseases [3,4,5,6].

Given that the autonomic nervous system (ANS) plays a relevant role in the pathogenesis of cardiovascular diseases, the promotion of interventions that aim to improve autonomic modulation of the cardiovascular system should be considered [6,7]. Physical exercise, owing to its beneficial effects on the cardiovascular system, is considered to be the cornerstone for the nonpharmacological treatment and prevention of such diseases in postmenopausal women [7,8,9,10,11]. Therefore, nonpharmacological treatment with different exercise methodologies, mainly aerobic, promotes, among other benefits, changes in cardiac autonomic balance, characterized by an increase in vagal autonomic drive and a decrease in sympathetic drive, associated with an increase in SBR [7,8,9,10,11,12,13,14,15].

Despite this, to the best of our knowledge, no systematic or updated analysis has described the effects of physical exercise on cardiovascular autonomic modulation in postmenopausal women. Therefore, this review aimed to investigate the effects of physical exercise on cardiovascular autonomic modulation in postmenopausal women.

## 2. Methods

A systematic review of the literature was performed by searching for randomized controlled trials (control or comparison arms) published between January 2011 and December 2021 in English that described the effects of physical exercise on the autonomic modulation of cardiovascular disease in postmenopausal women. Animal studies, cohorts that mixed pre- and postmenopausal states, women on chemotherapy or radiation therapy, and those that used combination treatment or showed acute effects were excluded from the analysis.

We searched for studies in the Scopus, Web of Science, PubMed, and Embase databases using search equations that included keywords extracted from Medical Subject Headings (MeSH) or EMTREE with Boolean descriptors “OR” within the word group and “AND” to combine terms related to population, intervention, control group, and outcomes.

The keywords used for the population were: postmenopause, postmenopausal female, postmenopausal period, and postmenopausal women; for the type of study: randomized clinical trial, controlled clinical trial, and controlled clinical comparison; for the intervention: exercise, exercise therapy, exercise training, and physical exercise; for the results: autonomic nervous system, heart rate, blood pressure, blood tension, arterial baroreflex, tilt-table test, heart-rate variability, and blood pressure variability.

In the study selection process, one author (J.C.S.-D.) eliminated duplicate records obtained through the search strategy. Then, two investigators (A.M.J.-H., K.Y.d.M.) screened the abstracts of potentially eligible studies and performed a full-text review to confirm that the eligibility criteria were met. The analyzed studies were developed independently by the evaluators and agreed upon by a third party (J.C.S.-D.) when there were discrepancies.

Subsequently, the methodological quality of each selected article was established using the PEDro scale (www.pedro.org.au, accessed on 16 January 2023) [16,17,18]. This scale comprises 11 items that assess eligibility, randomization, blinding, allocation masking, group comparability at the baseline, masking, intent-to-treat analysis, and adequate follow-up outcomes. The total score on the PEDro scale is the sum of each met criterion, ranging from 0 to 10. The higher the score is, the better the methodological quality of the study is. A summary score of 10–11 points was classified as excellent, 7–9 points as good, 5–6 points as fair, and less than 5 points as poor methodological quality [19]. The evidence quality or certainty was rated with the GRADEPro tool regarding the risk of bias, imprecision, inconsistency, indirectness, and publication bias.

Information related to the sample size, age, type of intervention, and results found to cardiovascular autonomic modulation (linear and nonlinear analyses of HRV and BPV) was extracted and analyzed.

The linear analysis of HRV included time domain and frequency domain parameters. RR interval (RRi), root mean square of the successive differences (RMSSD), standard deviation of NN intervals (SDNN), coefficient of variation (CV), number of pairs of successive NN (R–R), and intervals that differed by more than 50 ms (NN50) were the indices selected under the time domain. Total spectral power (TP ms2), high-frequency power in absolute (HF ms2) and normalized (HF nu) units, low-frequency power in absolute (LF ms2) and normalized (LF nu) values, the LF/HF ratio with absolute and normalized values (LFn/HFn), and geometric indices, such as the triangular index (RRtri), the triangular interpolation of RR, and interval histogram (TINN) were selected from the frequency domain HRV indices.

The analyzed nonlinear HRV parameters included the standard deviation of instantaneous beat-to-beat variability (SD1), the standard deviation of the long-term variability (SD2), the ratio of SD1/SD2, and detrended fluctuation analysis (DFA) (short-term fractal exponent α1, long-term fractal exponent 2, and the ratio of exponents α1/α2). BPV was obtained using the LF of the systolic blood pressure (SBP).

The synthesis and analysis of the information were narrative and qualitative. When possible, meta-analyses were performed using RevMan 5.4 to compare differences in means and 95% confidence intervals (95% CI) for continuous variables between the intervention and control/comparison groups. Lastly, TSA 0.9.5.10 Beta software was used to perform trial sequential analysis (TSA) on results associated with the meta-analyses. The parameters of this study were set as follows: Type I error probability α = 5%, and statistical power 1 − β = 90%. The sample size was used as the required information size (RIS) for a two-sided test. The protocol for this systematic review was registered with PROSPERO (CRD42022320414).

## 3. Results

The search criteria identified 91 studies, 8 of which met the eligibility criteria for analysis (Figure 1). These included samples of 25 to 64 physically inactive postmenopausal women, most apparently healthy [20,21,22,23], with obesity [24,25,26], hypertension [26], or coronary artery disease [27], whose ages ranged between 50 and 66 years. According to the PEDro scale, all studies specified eligibility criteria, used randomization, initial and intergroup comparison, and variability measures, and assessed at least one key outcome in more than 85% of the initially allocated subjects. None of the studies was blinded, nor was there any blinding of the participants, interventionists, or intention-to-treat analyses. The identified studies scored between 4 and 7 on the respective scales (Table 1). As to the quality of the evidence at the meta-analysis outcome level, Table 2 shows very low levels of certainty.

The analyzed studies investigated the effects of different physical training modalities on cardiovascular autonomic modulation assessed through HRV and BPV. The protocols included aerobic exercise in the home environment, functional training (FT), body vibration, resistance, and muscle stretching.

### 3.1. Effects of Physical Exercise on Linear Parameters of HRV

The effects of physical exercise on linear HRV parameters were evaluated in six studies, two of which showed significant changes in the time domain [20,21], while four showed changes in the frequency domain [21,22,23,24] (Table 3).

#### 3.1.1. Effects of Physical Exercise on HRV Time Domains

Two trials reported relevant results regarding the time domain. Rezende et al. [20] showed that 12 weeks of FT at a frequency of three times a week and intensity of 13–14 (Borg scale) increased the RRi (Control Group (Con) −22.66 ± 75.75 vs. Experimental Group (Exp) 70.17 ± 104.30, *p* < 0.05) and RMSSD (Con −0.18 ± 5.66 vs. Exp 5.10 ± 11.93, *p* < 0.05). Shen et al. [21] showed that physical exercise based on an aerobic modality can reduce the SDNN (Con −2.60 ± 1.81 vs. Exp −7.33 ± 2.15, *p* < 0,05), CV (Con −0.40 ± 0.17 vs. Exp −0.81 ± 0.23, *p* < 0.05) and the NN50 (Con 3.91 ± 1.87 vs. Exp −5.81 ± 2.27, *p* < 0.05) (Table 3).

#### 3.1.2. Effects of Physical Exercise on HRV Frequency Domains

Of the six trials in which linear analysis was performed in the frequency domain, four indicated the presence of significant changes in HRV. Lai et al. [27] reported that eight weeks of home aerobic exercise at a frequency of three times a week and intensity 13–15 (Borg scale), increased the TP (Con −107.24 ± 270.3 vs. Exp 165.25 ± 215,0; *p* < 0.05), HF ms^2^ (Con 3.07 ± 37.81 vs. Exp 91.0 ± 106.7; *p* < 0.05), and LF ms^2^ (Con −17.67 ± 55.83 vs. Exp 48.12 ± 73.32; *p* < 0.05). Wong et al. [24] showed that eight weeks of training with vibrating platforms, three times a week and with progressively increased intensity between 25–40 Hz, promoted a reduction in LnLF/LnHF ratio (Con −0,01 ± 0.02 vs. Exp −0.09 ± 0.0; *p* < 0.05), nLF/nHF (Con 0 ± 0.2 vs. Exp −0.3 ± 0.3; *p* < 0.05), and an increase in nHF (Con 2.1 ± 2.54 vs. Exp 8.2 ± 4.03; *p* < 0,05). Similarly, Wong et al. [25] observed that, after eight weeks of stretching exercises performed three times a week, there was a significant increase in the nHF (Con 1.1 ± 2.75 vs. Exp 13.9 ± 1.87 *p* < 0.05) and reduction in nLF (Con −2.8 ± 2.69 vs. Exp −14 ± 1.9; *p* < 0.05) and LnLF/LnHF (Con −0.01 ± 0.0 vs. Exp −0.08 ± 0.02; *p* < 0.05). Finally, Shen et al. [21] found that ten weeks of aerobic exercise performed using a step, three times a week, and with intensity between 75% and 85% of the reserve heart rate (RHR), promoted a reduction of 55.8% in LF, 39.9% in HF, 11.2% in normalized LF, and 34.5% in LF/HF ratio. In addition, an increase of 40% was observed in normalized HF (Table 3).

### 3.2. Effects of Physical Exercise on Nonlinear HRV Parameters

According to Rezende et al. [22], individuals who underwent FT for 18 weeks, three times a week, showed an increase in the SD1 component (Con −0.13 ± 4.0 vs. Exp 3.6 ± 8.43, *p* < 0.05) and in the fractal property of α1 (Con −0.04 ± 0.13 vs. Exp 0.07 ± 0.21) (Table 3). The meta-analysis revealed an increase in SD1 and SD2 in the trained group compared to the control (SD1: MD = 3.99; 95% [CI] = 1.22 to 6.77, n = 46; I^2^: 0%), (SD2: MD = 11.37; 95% CI = 2.99 to 19.75; n = 46; I^2^: 0%). (Figure 2A,B). Additionally, the TSA results show that the cumulative Z value crossed the traditional boundary value; however, the RIS curve was not crossed, indicating that the study sample size did not reach the expected value, as shown in Figure 3A,B.

### 3.3. Effects of Physical Exercise on BPV

The effects of physical training on BPV were investigated in only one of the evaluated studies [26]. The study showed that eight weeks of stretching exercises at a frequency of three times a week and with 50 min per session promoted a significant reduction in LFSBP in obese postmenopausal women (Con −0.05 ± 4.0 vs. Exp 1.62 ± 0.05, *p* < 0.001).

## 4. Discussion

The results of the review indicate that physical exercise may have beneficial effects on cardiovascular autonomic modulation in postmenopausal women, as reflected by an increase in the linear and nonlinear indices of HRV and BPV. However, the methodological quality of most of the studies was fair or poor, suggesting that the analyzed results may have been overestimated.

On the basis of studies that performed linear analysis of HRV, we observed that FT and aerobic training using a step caused an increase in RRi and RMSSD, and a reduction in SDNN, CV, and NN50. The mechanisms responsible for changes generated by FT are still unknown; however, the literature assumes that they may be partly related to the improvement of vascular endothelial function, resulting in an increased production of nitric oxide (NO), which is associated with stimulation and a lower concentration of renin after a physical training session, thereby inducing a lower amount of angiotensin II, which has an inhibitory effect on the cardiac vagal component [28]. However, the decrease in HRV parameters in the time domain caused by aerobic exercise using a step [21] had a discrepancy when compared to other studies carried out in postmenopausal women [29,30]. This finding can be explained by the monitoring period used, noting that the total variance of HRV increased with the duration of the recording [31].

Regarding the frequency domain, there was an increase in the vagal modulation measured by the HF component in normalized units, and a decrease in the sympathetic modulation measured by the LF component in absolute units and LF/HF ratio after different training modalities, such as aerobics at home, stretching exercises, aerobics using a step, and a vibrating platform [21,24,25,27]. In addition to the increase in NO synthesis, the decreases in circulating angiotensin II and arterial stiffness are possible responsible factors for the vagal predominance after treatment with these four modalities [28,32,33,34]. There is another hypothesis that could specifically explain the effect of stretching on autonomic modulation, which can generate a state of mental–physical hypometabolic relaxation that promotes a hypothalamic response characterized by a decrease in cardiac sympathetic autonomic influence [35].

Our study suggests that both aerobic and strength training can increase the SD1, SD2, and short-term fractal (α1; α1/α2) indices. These findings are important considering that nonlinear methods have shown a new view of HRV behavior in different conditions, providing additional prognostic information on cardiovascular morbidity/mortality when compared to linear methods [3,4,36,37,38]. On the basis of the TSA, more studies are necessary to confirm the effectiveness of this type of training on the behavior of these HRV parameters in postmenopausal women.

In turn, BPV was analyzed only in a single study, and was significantly reduced in postmenopausal women undergoing eight weeks of stretching exercises [25]. Resulting from the sum of the mechanisms that respond to disturbances external to the organism, this variable can be classified according to the time interval in which it is evaluated. When analyzed in short beat-to-beat intervals, the method used in this work, the observed variation was mainly associated with the sympathetic modulation of the vascular tone [39,40]. Although population studies are still needed to enhance the clinical validity of this method, some studies have suggested that high values are associated with a higher risk of cardiovascular outcomes and mortality. Furthermore, changes in the respective parameters may be indicative of changes in vascular sympathetic activity. Although few studies have evaluated the effects of stretching exercises on vascular sympathetic modulation, the literature suggests that, when performed chronically, this modality can improve endothelial function by reducing vessel stiffness and blood pressure. The explanation for such effects also involves the increase in NO bioavailability promoted by this type of activity [41]. In postmenopausal women, musculoskeletal discomfort and physical limitations may be frequent, which are related to low motivation to adhere to conventional exercise programs, especially high-intensity ones [42,43]. This is the reason why nontraditional modalities such as those developed at home, and vibration and stretching training could be considered to be good alternatives to improve cardiovascular autonomic balance, especially in the early stages of treatment. Other possible advantages are the relatively low cost and easy applicability, which would allow for training to be carried out even at home.

There are some limitations to our findings that should be considered when interpreting the results: the small number of randomized clinical trials, with few participants, the lack of control for confounding factors (dietary habits, the use of drugs or hormone therapy, and morbidity status) [22,27], and the poor methodological quality determined by the very nature of the studies that does not allow for the use of placebo or of analysis selection by intervention protocol and not by intention to treat, which can generate a bias in the results [44]. Lastly, it is suggested to include the processes of assessment of BRS, and levels of catecholamines and angiotensin II, which could contribute to a more comprehensive understanding of the effects of exercise on cardiovascular autonomic modulation in postmenopausal women.

## 5. Conclusions

According to the analyzed studies, aerobic physical exercise and some unconventional training modalities could have beneficial effects on cardiovascular autonomic modulation in postmenopausal women. However, considering the limitations of this study, these results should be interpreted with caution. New clinical trials with larger samples and greater methodological rigor that compare alternative and conventional treatments in longer intervention periods would also be useful.

## Figures and Tables

**Figure 1 ijerph-20-02207-f001:**
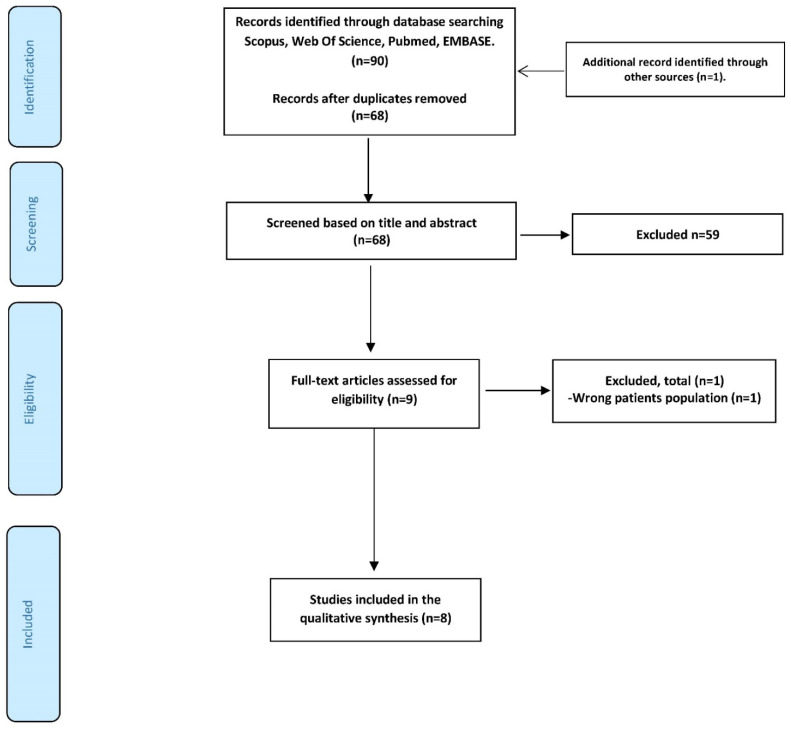
Methodological quality for the included trials.

**Figure 2 ijerph-20-02207-f002:**
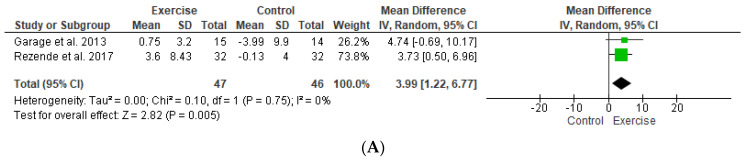
Forest plot of meta-analysis results presented as pooled standard mean differences with 95% CIs for changes in (**A**) SD1 and (**B**) SD2 for exercise and control groups. The exercise effects are plotted with black diamonds. IV, inverse variance; SD1, standard deviation of instantaneous beat-to-beat variability; SD2, standard deviation of the long-term variability [22,23].

**Figure 3 ijerph-20-02207-f003:**
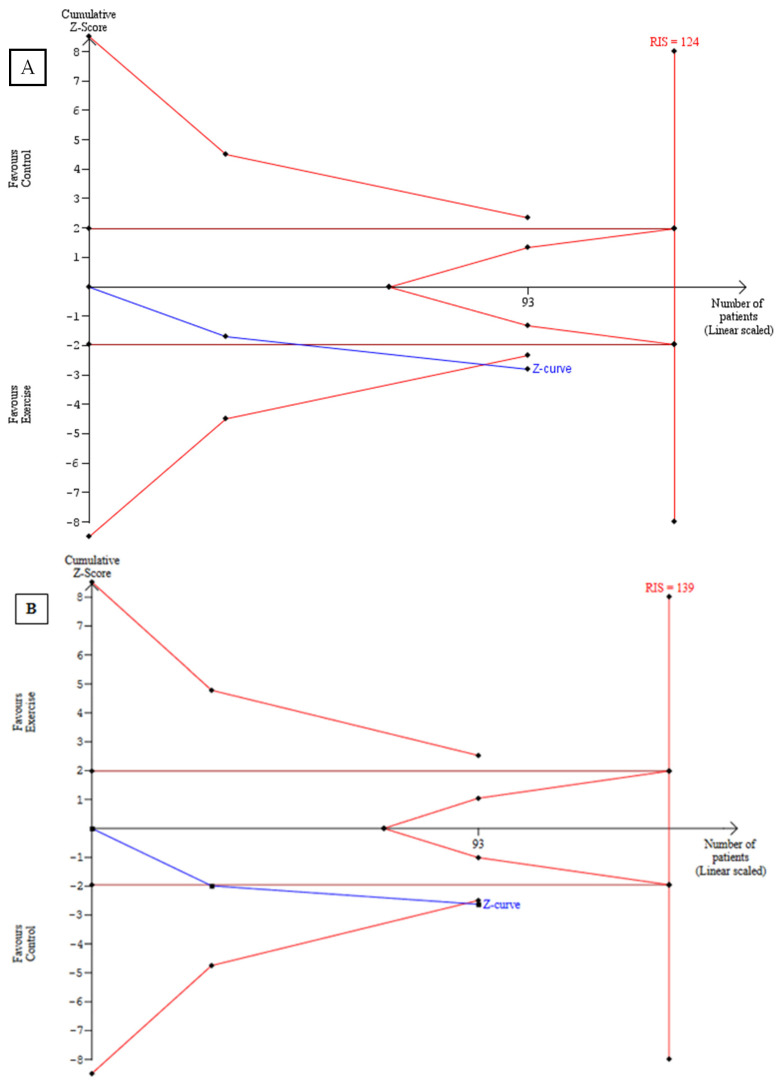
Trial sequential analysis—effect on (**A**) SD1 and (**B**) SD2.

**Table 1 ijerph-20-02207-t001:** Methodological quality for the included trials.

	Methodological Quality (PEDro Scale)												
First Author	Eligibility Criteria	Random Assignment	Allocation Concealment	Initial Comparability	Participant Blinding	Provider Blinding	Blinding of Outcome Assessors	Retention	Analysis by Intention to Treat	Comparison between Groups	Variability of Key Outcome Provided	Total Score	Quality of Study
Rezende et al. [20]	Y	Y	N	Y	N	N	N	N	N	Y	Y	4	Poor
Shen et al. [21]	Y	Y	N	Y	N	N	N	N	N	Y	Y	4	Poor
Rezende et al. [22]	Y	Y	N	Y	N	N	N	N	N	Y	Y	4	Poor
Gerage et al. [23]	Y	Y	N	Y	N	N	N	Y	N	Y	Y	5	Fair
Wong et al. [24]	Y	Y	N	Y	N	N	N	N	N	Y	Y	4	Poor
Wong et al. [25]	Y	Y	Y	Y	N	N	Y	N	N	Y	Y	7	Good
Wong et al. [26]	Y	Y	N	Y	N	N	N	Y	N	Y	Y	6	Fair
Lai et al. [27]	Y	Y	N	Y	N	N	N	N	N	Y	Y	4	Poor

Abbreviations: N, no; Y, yes.

**Table 2 ijerph-20-02207-t002:** Summary of findings and certainty assessment–intervention (exercise vs. control).

Certainty Assessment							Number of Patients		Effect		Certainty	Importance
№ of Studies	Study Design	Risk of Bias	Inconsistency	Indirectness	Imprecision	Other Considerations	Exercise	Control	Relative (95% CI)	Absolute (95% CI)		
**Instantaneous beat-to-beat variability (follow-up: mean 14 weeks)**												
2	randomised trials	very serious ^a^	not serious	not serious	serious ^b^	none	47	46	-	3.99 (1.22 to 6.77)	⊕◯◯◯ Very low	No importante
**Standard deviation of the long-term variability (follow-up: mean 14 weeks)**												
2	randomised trials	very serious ^a^	not serious	not serious	serious ^b^	none	47	46	-	11.37 (2.99 to 19.75)	⊕◯◯◯ Very low	No importante

CI: confidence interval; MD: mean difference. Explanations: ^a^ High risk of reporting bias in trials with >25% weight. ^b^ Total population size or number of events is less than 400.

**Table 3 ijerph-20-02207-t003:** Effect of physical exercise and HRT on cardiovascular and metabolic system in postmenopausal women.

Study		Participants *n* at Baseline	Age	Groups	Evaluated Outcomes	Significant Differences between Groups
Lai et al. [27] (2011)	CON	*n* = 19	66.69 ± 5.26	Control group did not participate in any supervised exercise.	Linear HRV indexes. Time domain: SDRR. Frequency domain: TP (ms^2^), LF (ms^2^), HF (ms^2^), LF (nu), HF (nu), LF/HF.	A home-based exercise program appears able to improve HRV: (TP, HF, LF).
	EX	*n* = 21	64.19 ± 5.86	Type: Aerobic—at home Each session consisted of a 5 min warm-up of slow walking, a brisk-walking exercise segment of 30 min, and a 5 min cooldown of slow walking. D: 8; Ds: 40; F: 3; I: 13 and 15 on the Borg scale during the fast walk.		
Wong et al. [23] (2016)	CON	*n* = 12	59 ± 1	Control group did not participate in any supervised exercise.	Linear HRV indexes. Time domain: LnRMSSD. Frequency domain: LnTP ms, nLF, nHF, nLF/nHF, LnLF ms^2^, LnHF ms^2^, LnLF/LnHF ms^2^.	LnLF/LnHF and nLF/nHF decreased, and nHF increased after WBV (*p* ≤ 0.01). The control group showed no changes.
	EX	*n* = 13	58 ± 1	Type: Whole-Body Vibration: Subjects performed four static and four dynamic leg exercises on a vibration platform. Training intensity progressed increasing the vibration (25–40 Hz) and the frequency from low to high amplitude (4, 5 and 21.3 gr). D: 8; Ds: 30–60; F: 3 separated by at least 48 h.		
Wong et al. [24] (2017)	CON	*n* = 12	58 ± 1	Control group did not participate in any supervised exercise. Participants were instructed not to change their regular habits during the study.	Linear HRV indexes. Time domain: LnRMSSD, ms Frequency domain: LnTP ms, nLF, nHF, nLF/nHF, LnLF ms^2^, LnHF ms^2^, LnLF/LnHF ms^2^	For the Stretching training (ST) group between baseline and post intervention, there were significant increases in parasympathetic modulation as measured by (nHF), *p* < 0.01, and decreases in sympathetic modulation as measured by nLF and LnLF/LnHF (<0.05).
	EX	*n* = 13	57 ± 1	Type: Stretch training Supervised sessions with one set of 18 active and 20 passive stretches (20 exercises in a standing position, 8 in a sitting position, and 10 in lying position). A stretched muscle was held for 30 s at the point of maximal exertion or range of motion. Each stretch was followed by a 15 s relaxation period. D: 8; Ds: 60; F: 3		
Shen et al. [21] (2013)	CON	*n* = 30	59.1 ± 0.83	Control group did not participate in any supervised exercise and were asked not to change their physical activity habits during the study.	Linear HRV indices. Time domain: RR mean, SDNN (ms), CV, NN50, pNN50, NN20, pNN20, RMSSD SDSD Frequency domain: TP (ms^2^), VLF (ms^2^), nHF, nLF, LF/HF.	The SAE had significant mean decreases in SDNN, CV, NN50, LF (ms^2^), HF (ms^2^), nLF, and LF/HF, and showed a significant increase in nHF.
	EX	*n* = 32	57.86 ± 0.64	Type: Step-aerobic exercise SAE. Each session: (1) warm-up: 10–15 min stretching, (2) SAE: 35–40 min, (3) balance and cool down: 10–15 min, and (4) stretching and relaxation: 10–15 min. The step aerobic exercise program was performed in the low-impact version in which one foot remains in contact with the ground or bench at all times, preventing any hopping or jumping movements. The music cadence of all sessions was set between 120 and 126 foot strikes per minute. D: 10; Ds: 90; F: 3; I: 75–85% HRR.		
Rezende et al. [20] (2019)	CON	*n* = 32	58.45 ± 4.8	Control group did not participate in any supervised exercise.	Linear HRV indexes. Time domain: RR interval, SDNN, RMSSD ms Frequency domain: nLF, nHF, LF ms^2^, HF ms^2^	The results obtained from the training showed improvement of the following cardiac parameters in the FTG: heart rate, RR intervals and RMSSD.
	EX	*n* = 32	60 ± 4.5	Type: Functional Training. Eleven stations, passing three times in each station with a rest time of thirty seconds. When finished, the participant spends 18 to 30 min walking. D: 18; F: 3; I: 13–14 Borg scale.		
Rezende et al. [22] (2017)	CON	*n* = 32	58.45 ± 4.8	Control group did not participate in any supervised exercise.	Linear HRV indexes. RRTri, TINN. Nonlinear HRV indexes. SD1, SD2, SD1/SD2, DFA (α1, α2, α1/α2).	The trained subjects had increased SD1, beat-to-beat global dispersion much greater, increased in the dispersion of long-term RR intervals and increased short-term fractal properties (α1).
	EX	*n* = 32	50 ± 4.5	Type: Functional Training. Eleven stations, passing three times in each station with a rest time of thirty seconds. When finished, the participant spends 14 to 30 min walking. D: 18; F: 3		
Wong et al. [26] (2014)	CON	*n* = 14	57 ± 1	Control group did not participate in any supervised exercise. Participants were instructed not to make changes in their regular habits during the study.	Blood pressure variability: LFSBP	ST reduces vascular sympathetic activity in obese postmenopausal women with prehypertension and hypertension.
	EX	*n* = 14	56 ± 1	Type: Stretch training. Supervised sessions, with one set of 18 active and 20 passive stretches (20 exercises in a standing position, 8 in a sitting position, and 10 in the lying position). A stretched muscle was held for 30 s at the point of maximal exertion, or range of motion. Each stretch was followed by a 15 s relaxation period. D: 8; Ds: 60; F: 3		
Gerage et al. [23] (2013)	CON	*n* = 14	66.2 ± 4.1	Control group did not participate in any supervised exercise.	Linear HRV indexes. Time domain: RR interval, HR (bpm), SDNN, RMSSD. Frequency domain: HF (ms^2^), LF (ms^2^), lnLF(ms^2^), lnHF(ms^2^), LF/HF. Geometric index: RRTRI, TINN. Nonlinear HRV indexes. SD1, SD2, SD1/SD2	The results of our study suggest that a 12-week supervised RT program does not affect HRV in postmenopausal, nonhypertensive, untrained elderly women.
	EX	*n* = 15	65.5 ± 5	Type: Resistance training. The RT program was a total body program with 8 exercises (machine bench press, leg extension, wide-grip front lat pulldown, leg curl, preacher curl seated calf raise, triceps pushdown, and abdominal crunches); 2 sets, 10–15 repetitions. D: 12; F: 3; I: 13–14 Borg scale. Subjects performed 2 consecutive sets of 10–15 repetitions until moderate fatigue in each exercise or stopped when it began to be difficult. The only exception was the abdominal crunch exercise which was performed on the floor using the subject’s body weight (20 to 30 repetitions without any additional overload)		

*n*, sample size at baseline;; CON, control; EX, exercise; D, duration of the intervention (weeks); Ds, duration of the exercise session (minutes); F, exercise frequency (times/week); WBV, whole-body vibration; Ln, natural logarithm; TP, total power; LF, low frequency; HF, high frequency; LF/HF, LF to HF ratio; n, normalized unit; CV, coefficient of variation; CAV, ; SDRR, SD of the RR; RMSSD, root mean square of successive differences of R–R interval; RRTri, triangular index; TINN, triangular interpolation of RR intervals SD1, standard deviation of instantaneous beat-to-beat variability; SD2, standard deviation of the long-term variability; SD1/SD2, ratio between SD1/SD2; DFA, detrended fluctuations analysis; α1, short-term fractal exponent; α2, long-term fractal exponent; α1/α2, ratio between the exponents; LFSBP, low-frequency component of systolic blood pressure; RT. resistance training; HRR, heart rate reserve.

## Data Availability

The extracted data used to support the findings of this study are available from the corresponding author upon request.

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
