# Peer review of "Physical Exercise Effects on Cardiovascular Autonomic Modulation in Postmenopausal Women—A Systematic Review and Meta-Analysis"

_ijerph, 2023, doi:10.3390/ijerph20032207_

Round 1

Reviewer 1 Report

Dear authors, 

The present article investigated the effects of physical exercise on cardiovascular autonomic modulation in a postmenopausal woman as a meta-analysis. The inclusion of these items in the meta-analysis is a novel point with respect to previous articles. Furthermore, it allows a better understanding of the possible interactions, which makes this research relevant. Next, I detailed some aspects of your manuscript that should be corrected or improved.

Kind regards,

Author Response

Revise and Resubmit.

January 16, 2022

Dear Editor,

We would like to thank the Editor and the reviewers for their thoughtful comments on our paper. They were very helpful in improving the quality of our manuscript. Replies to the questions raised by the reviewers and a list of all the changes made are presented below.

Review #1

  1. Line 7: Abstract. There is no background but only an explanation of the study's aim. Please add more background.

Response: We added the background.

  1. Line 9 or 50: Studies regarding this topic were limited from January 2011 to December 2021. There appear to be eight papers that fit the current criteria. I want to ask that there is sufficient reason for limiting this to this period. Please explain the reasons that it fits in explaining this purpose. Among the ten years of research, only eight studies are included, and mainly two are used. So, it needs additional explanations related to the investigation and reasons because it is difficult to prove the value of this study.

Response:

The search is limited to the last 10 years due to:

- When performing the search without including the publication time filter, most of the articles were published in the time frame used (2011-2021). In addition, most of the articles published before these dates showed short-term effects and or combined exercise with other treatments (exclusion criteria).

- We would like to present the most up-to-date evidence on this topic.

  1. Line 25: This paper started talking about menopause and cardiovascular disease. In this regard, additional explanation is needed in the introductory part about why exercise is necessary for postmenopausal women.

Response: We appreciate the suggestion. However, we believe that the importance of physical exercise is described in the penultimate paragraph (Page 2, paragraph 1).

  1. Line 48: Eight papers were finally adopted after excluding articles screened through the method. In this regard, please explain the progress in the method section according to the exclusion criteria of Figure 1.

Response: Thank you for this suggestion. We edited the text to clarify the article selection process (Page 2, paragraph 4 and 5).

  1. Line 99 - Line 115: I recommend that it is explained the following related parts in the method section. It is expected that future reviewers will be able to understand easily by classifying and defining the analyzed items according to the results.

Response: Following the suggestions of the reviewer, is explained in the method section the classifying and defining the analyzed items according to the results (Page 2, paragraph 9; Page 3, paragraph 1 and 2).

  1. Line 97: Table 1. "Variation of key outcome provided" is truncated.

Response: Table 1 adjusted.

  1. Line 116: Result 3.1 effects of physical exercise on linear parameters of HRV. There is no result value expression. It was expressed, "The results are as follows": I wonder if the result should come next. If the results are the same as the below "results," it needs to be an additional explanation.

Response: the titles of the sub-subsection are adjusted to give clarity to the results of the linear parameters of HRV presented (Page 5).

  1. Line 85: Results show the data value in writing, but it seems necessary to tell what tables or figures to help understand the form, like "result 3.4 Effects of physical exercise on non-linear HRV parameters."

Response: Thanks for your observation. We added the reference in every paragraph indicating the tables and figures that help to understand the results described.

Reviewer 2 Report

1. Introduction can be made smaller and more emphatic 

2. I see the authors didnt observe much heterogenity in the metanlysis can the authors use trial sequential analysis to forsee the same to estimate whether we are not over estimating the data. 

3. Did the authors do Risk of Bias assessment ?

4. Most of the included trials were of small sample with no information on Co-Morbid status ? were all the participants healthy ?

5. Discussion and conclusion to be readdressed incase the desired Statistics are done. 

Author Response

Revise and Resubmit.

January 16, 2022

Dear Editor,

We would like to thank the Editor and the reviewers for their thoughtful comments on our paper. They were very helpful in improving the quality of our manuscript. Replies to the questions raised by the reviewers and a list of all the changes made are presented below.

Review #2

  1. Introduction can be made smaller and more emphatic 

Response: We appreciate the suggestion. However, we consider that the introduction section is brief and complete. This means if we try to make it smaller we would need to remove relevant information that we consider necessary for the readers.

  1. I see the authors didn’t observe much heterogenity in the metanlysis can the authors use trial sequential analysis to forsee the same to estimate whether we are not over estimating the data. 

Response: Following the reviewer's suggestions, the trial sequential analysis was performed. (Page 3, paragraph 3; Page 6, paragraph 2; Page 9, paragraph 4).

  1. Did the authors do Risk of Bias assessment?

Response: Thanks for the observation. In Table 2 the methodological quality of the results of the meta-analysis qualified with the GRADEPro tool includes the risk of bias analysis (Page 2, paragraph 8. See table 2S).

  1. Most of the included trials were of small sample with no information on Co-Morbid status? were all the participants healthy ?

Response: Thanks for the observation. We modified the text specifying that not all women were healthy. This was one of the limitations to consider when interpreting the results. (Page 3, paragraph 4).

  1. Discussion and conclusion to be readdressed incase the desired Statistics are done. 

Response: The discussion and conclusion were structured according to the suggestions.

Round 2

Reviewer 2 Report

With the desired changes the manuscript appears acceptable